# Kompetitive Allele Specific PCR Genotyping of 89 SNPs in Romanian Spotted and Romanian Brown Cattle Breeds and Their Association with Clinical Mastitis

**DOI:** 10.3390/ani13091484

**Published:** 2023-04-27

**Authors:** Daniela Elena Ilie, Dinu Gavojdian, Szilvia Kusza, Radu Ionel Neamț, Alexandru Eugeniu Mizeranschi, Ciprian Valentin Mihali, Ludovic Toma Cziszter

**Affiliations:** 1The Research Department, Research and Development Station for Bovine Arad, 310059 Arad, Romania; 2The Research Department, Research and Development Institute for Bovine Balotesti, 077015 Balotesti, Romania; 3Centre for Agricultural Genomics and Biotechnology, University of Debrecen, 4032 Debrecen, Hungary; 4Department of Life Sciences, Faculty of Medicine, “Vasile Goldiș” Western University of Arad, 310025 Arad, Romania; 5Department of Animal Production Engineering, Faculty of Bioengineering of Animal Resources, University of Life Sciences ‘King Mihai I’ from Timișoara, 300645 Timișoara, Romania

**Keywords:** KASP, mastitis, polymorphism, Romanian Brown, Romanian Spotted

## Abstract

**Simple Summary:**

In order to improve animals’ resistance to mastitis in subsequent generations and to reduce disease prevalence, one feasible approach could be the genomic selection of cows with the most potentially protective immune responses. In this study, 298 cattle were genotyped for 89 single nucleotide polymorphisms through the Kompetitive Allele Specific PCR. Our findings reveal a significant association of a molecular marker in the *BOLA-DRB3* gene with mastitis prevalence in the investigated breeds. This supports previous studies that the *BOLA-DRB3* gene represents an important locus affecting resistance to mastitis in cattle. These results contribute to an increase in knowledge and available data regarding the genetic variability of SNPs for udder health in cattle.

**Abstract:**

Mastitis is the most common production disease in the dairy sector worldwide, its incidence being associated with both cows’ exposure to bacteria and the cows’ genetic make-up for resistance to pathogens. The objective of our study was to analyse 89 missense SNPs belonging to six genes (*CXCR2, CXCL8, TLR4, BRCA1, LTF, BOLA-DRB3*), which were found to be associated with genetic resistance or susceptibility to mastitis. A total of 298 cattle (250 Romanian Spotted and 48 Romanian Brown) were genotyped by Kompetitive Allele Specific PCR (KASP) and a chi-squared test was used for genetic association studies with clinical mastitis. A total of 35 SNPs (39.3%) among the selected 89 SNPs were successfully genotyped, of which 31 markers were monomorphic. The polymorphic markers were found in two genes: *TLR4* (rs460053411) and *BOLA-DRB3* (rs42309897, rs208816121, rs110124025). The polymorphic SNPs with MAF > 5% and call rates > 95% were used for the association study. The results showed that rs110124025 in the *BOLA-DRB3* gene was significantly associated with mastitis prevalence (*p* ≤ 0.05) in both investigated breeds. Current results show that the SNP rs110124025 in the *BOLA-DRB3* gene can be used as a candidate genetic marker in selection for mastitis resistance in Romanian dairy cattle.

## 1. Introduction

Globally, mastitis is a prevalent disease that impairs animal health and welfare, resulting in serious economic losses [1,2]. In cattle, it is one of the most significant health issues, along with infertility and lameness [3]. In expectation of current customer demands for dairy products with high welfare standard certifications, the dairy sector should increase concern for improving the welfare of dairy cattle and, thus, mastitis prevention and control should be of equal concern to farmers, veterinarians and researchers.

Intense selection in recent decades in dairy cows and the use of elite sires to maximize milk yields has had detrimental effects on other functional traits, such as the incidence of clinical mastitis [4,5]. At the same time, the recent decade’s scientific advances in molecular genetics, followed by the sequencing of the *Bos taurus* genome, provided the opportunity to identify single nucleotide polymorphisms (SNPs) in various genes that could possibly be utilized as genetic markers for resistance or susceptibility to mastitis. Researchers and livestock breeders are therefore trying to utilize genetic variation in resistance to mastitis to manage this pressing issue [6]. Many candidate genes or polymorphisms within these genes have been identified that are associated with mastitis in dairy cattle [6,7,8,9,10,11,12,13]. However, since resistance to this condition is a complex multifactorial trait, more work needs to be done to fully understand the genetic determinants, as well as to find new markers that will enable a more rigorous selection. Therefore, the analysis of markers in genes already known to be associated with udder health is a straightforward approach that does not require a large number of animals to detect SNPs associated with traits of interest, as in the case of genome-wide association studies [6].

In accordance with other studies carried out so far, for the present study we have chosen six genes that have been previously associated with mastitis. The lactotransferrin (*LTF*) gene maps to *Bos taurus* autosome (BTA) 22, which is a member of the transferrin gene family, consists of 17 exons and 2722 variant alleles, has an antibacterial, antiviral, antifungal and anti-inflammatory activity [14] and is regarded as a candidate gene for increasing resistance against infections of the mammary gland in dairy cattle [7,8]. The toll-like receptor 4 (*TLR4*) gene maps to BTA8, consists of four exons and the protein encoded by this gene is a member of the toll-like receptor (TLR) family that plays an important role in pathogen recognition and activation of innate immunity [15]. This gene has previously been considered as a genetic marker for mastitis resistance in dairy cows [9,10,11]. The *IL8*/*CXCL8* (C-X-C motif chemokine ligand 8) gene maps to BTA6 and the protein encoded by this gene, which is referred to as interleukin-8 (IL-8), is an important mediator of the inflammatory response and acts on the CXCR1 and CXCR2 receptors [16]. The *CXCR2* (C-X-C motif chemokine receptor 2) gene is located on BTA2 and encodes the cytokine receptors necessary for neutrophil migration to infection sites [6] and different SNPs on this gene have been associated with susceptibility to inflammatory diseases, including subclinical mastitis in dairy cows [12]. The *BRCA1* (breast cancer 1) gene plays a crucial function in DNA repair of double-stranded breaks and in maintaining genomic stability [13]. This gene maps on to BTA19 with a total of 71.081 kb consisting of 22 exons, encodes a protein of 1849 amino acids and is associated with 2399 variant alleles. The *BOLA-DRB3* (major histocompatibility complex, class II, DRB3) gene is located on BTA23 and consists of six exons, of which the second exon is the most polymorphic. This gene plays a central role in the immune system and resistance to pathogens [17], being associated with differences in susceptibility to infectious diseases, somatic cell count and mastitis incidence.

The Romanian Spotted cattle (RS, national name Bălțată Românească) belongs to the Simmental strain, being a dual-purpose breed, with a current census of 376,000 cows, representing 36% of the breed structure in Romania [18]. The RS originates in the 18th century, being the result of non-systematic crossbreeding between Simmental bulls imported from Austria and Switzerland and local unimproved Podolic Grey cattle [19]. The average milk yield for the RS breed ranges between 5000 and 5700 kg of milk/lactation, adult body weight in cows is 600 to 620 kg and average daily gain in fattening young bulls is 1000 to 1200 g [20]. The selection index for the RS breed is focused on milk yield (50%), growth rates and carcass attributes (20%) and fitness related traits (30%) [21].

The Romanian Brown cattle (RB, national name Brună de Maramureș) belongs to the Braunvieh strain, having a census of 220,000 cows. The RB is the result of crossbreeding between the Braunvieh breed from Switzerland and local unimproved Mocănița and Grey cattle [19], with the herd-book being established in 1959. The average milk yield for the RB breed is highly variable, ranging between 3000 and 5000 kg of milk/lactation [22], the breed being reared mainly under pasture-based production systems at altitudes higher than 400 m. The selection programme for the breed is currently focused on improving the milk yield up to 5500 kg and increasing the body weight of adult cows to 600 kg.

Genomic selection for both RS and RB breeds has been introduced in the past five years in Romania, being used particularly in young bulls that are produced for AI stations. Although performance recordings and selection programmes are in place for both breeds, semen imports are occurring regularly in order to manage inbreeding rates and genetic gain.

Having in mind the importance of udder health, this study aimed to analyse the polymorphism of 89 SNP markers belonging to six genes that were found to be associated with genetic resistance to mastitis through the use of the Kompetitive Allele Specific PCR technique in Romanian Spotted and Romanian Brown cattle breeds.

## 2. Materials and Methods

### 2.1. Ethics Approval

The experimental design, sample collection protocols and procedures were approved by the Institutional Ethics Committee of the Research and Development Station for Bovine Arad belonging to the Academy for Agricultural and Forestry Sciences (Decision no. 51/ PN-II-RU-TE-1402). Blood samples were collected from animals by trained veterinarians. All research activities involved in the present study were performed in accordance with the Directive 2010/63/EU on the protection of animals used for scientific purposes.

### 2.2. Animals and General Management

A total of 298 cattle reared at the Research and Development Station for Bovine Arad (46°10′16.2″ N 21°14′08.6″ E) in Romania, including 250 Romanian Spotted and 48 Romanian Brown, were included in the research herd. The cows were both primiparous and multiparous, and the entire herd was included in the Official Performance and Recording Scheme.

Cows were milked twice per day in a ‘herringbone’ milking parlour (2 sides × 14 units), which employs the AfiFarm 5.4^®^ farm management software. Animals were fitted with AfiTag^®^ pedometers (Afimilk Ltd., Kibbutz Afikim, Israel). During the study, cattle were kept on deep straw bedding, with a space allowance of 9 m^2^ in the resting area and free access to outside paddocks, benefiting from a dry and clean environment. Cows from both breeds were housed in groups of 40 to 50 animals, according to lactation stage and productivity. Additionally, all cows had a feeding space allowance of about 75 cm/head, were fed twice per day and had unlimited access to water. The feeding of cows differed according to season. Daily individual diet consisted of 6 kg alfalfa hay, 6 kg concentrates and 35 kg maize silage during the winter and in the summer the silage was reduced to 18 kg and 45 kg of green fodder was added.

### 2.3. Data Collection

Milk yield per milking session and milk conductivity were recorded daily for a period of 20 months. Production levels and milk quality data were taken from the results of the Official Performance Control service. The individual milk samples were collected in the herd at intervals of 28 days in order to monitor udder health using somatic cell count (SCC) as a proxy for mastitis. A mastitis diagnostic was based on daily clinical observation of the udder during the 20-month trial. Clinical mastitis was initially identified by trained technicians on the basis of signs such as udder swelling, hardness of the affected quarter, abnormal milk and/or changes in the consistency or colour of the milk. In addition, data based on clinical signs were corroborated with SCC, where we used a threshold value of 285,000 somatic cells/mL of milk or greater. Only the cows that cumulatively met the two conditions were declared positive for clinical mastitis.

### 2.4. Selection of SNPs

The decision of which single nucleotide polymorphism (SNPs) to be used in the present study for genotyping was conducted gradually, with the first step being the candidate gene selection, followed by SNP identification and selection. A total of 89 bi-allelic SNPs were chosen across the cattle genome, based on previous results from the literature [12,23,24,25]. The selected SNPs belong to six genes associated with mastitis: *LTF* (lactotransferrin), *TLR3* (toll-like receptor 4), *IL8*/*CXCL8* (C-X-C motif chemokine ligand 8), *BRCA1* (breast cancer 1), *BOLA-DRB3* (major histocompatibility complex, class II, DRB3) and *CXCR2* (C-X-C motif chemokine receptor 2). The SNPs from the investigated candidate genes were collected by querying the Single Nucleotide Polymorphism Database maintained by the National Center for Biotechnology Information (NCBI) and Ensembl. The SNPs were generally selected to fall within the coding region of the gene and change the resulting codon (missense mutations). As a result, the final list of SNPs used in the study included 86 missense (non-synonymous) and 3 synonymous SNPs. Details regarding the SNP ID, gene symbol, chromosome position based on the ARS-UCD1.2 genome assembly of *Bos taurus*, alleles, synonymous/non-synonymous nature, position in coding region and amino acid change are presented in Table 1.

### 2.5. DNA Extraction and Genotyping

The bovine genomic DNA was extracted from total blood. Fresh blood (2 mL) was collected from the tail vein (vena cava) of the animals in vacutainers containing K3EDTA as an anticoagulant (Vacutest Kima, Padova, Italy). Genomic DNA was isolated from 300 µL blood samples using a commercial DNA extraction kit (Wizard Genomic DNA Purification Kit, Promega, Madison, WI, USA), according to the manufacturer’s instructions. After isolation, DNA concentration and quality were assessed using a NanoDrop 2000 Spectrophotometer (Thermo Fisher Scientific, Wilmington, DE, USA). Samples were subsequently diluted to a uniform concentration of 20 ng of DNA per sample for use in Kompetitive Allele Specific PCR (KASP) assays. All investigated DNA samples were sent to LGC Genomics (Teddington, Middlesex, UK) in order to perform the KASP genotyping for the bi-allelic discrimination of the panel of the 89 selected SNPs. Genotyping data were visualized as a cluster plot using the SNP Viewer software (version 1.99, Hoddesdon, UK).

### 2.6. Statistical Analysis

In order to identify the SNP genotypes, the raw allele call received from LGC Genomics was analysed using the KlusterCaller software from LGC Genomics. Considering that the aim of the present research was to investigate molecular markers for their use in designing future breeding programs, we conducted the analysis in order to see if polymorphic SNPs are suitable for the planned work by measuring the markers’ informativeness. Thereby, heterozygosity and the polymorphism information content (PIC) parameters were estimated. POPGENE 1.32 [26] was used to calculate the frequencies of alleles and genotypes, deviation from Hardy–Weinberg equilibrium (HWE), observed (Ho) and expected (He) heterozygosity values and PIC for each breed. A genetic association study was further conducted on polymorphic SNPs. In order to detect possible associations between the phenotypes and genotypes for all polymorphic SNPs, the chi-square test of independence was used, employing the statistical software Statistica [27]. Thus, each genotype within a SNP was associated with a phenotype measured at the animal level for mastitis frequency during the study period and the number of affected quarters when clinical mastitis was diagnosed. For mastitis frequency, three levels of phenotype were observed: no mastitis during the lactation, mastitis occurring once and mastitis occurring twice during a lactation. For the number of affected quarters, four levels of phenotype were observed: none of the quarters (associated with no mastitis), one quarter, two quarters and all four quarters. Markers with effect were considered those SNPs for which the chi-squared test showed a significant (*p* < 0.05) difference in genotype distribution within the same SNP.

## 3. Results

The present study was focused on analysing SNPs located in genes that have been previously found to be associated with genetic resistance to mastitis. Eighty-nine SNPs were analysed in 298 animals from the two breeds: Romanian Spotted (*n* = 250) and Romanian Brown (*n* = 48). The KASP genotyping method was used to analyse the SNPs across the six genes. A total of 35 SNPs (39.3%) among the selected 89 SNPs were successfully genotyped across the two breeds (Table 2).

Although the set of SNPs was selected from dbSNP, a large number of SNPs (*n* = 54, 60.7%) could not be genotyped through the KASP genotyping method. The following 31 SNPs were fixed: rs480434223, rs209319366, rs464576793, rs467207447, rs446793067 in *CXCR2*; rs134178216, rs468955241, rs448378725, rs466813222, rs453466519, rs452742998, rs445886432 in *CXCL8*; rs8193041, rs8193048, rs8193049, rs8193050, rs8193053, rs8193055, rs135659119, rs8193066 in *TLR4*; rs437880924, rs470945089, rs480362322, rs469103334 in *BRCA1*; rs460348489, rs461145277, rs476824727, rs447239896 in *LTF* and rs208667846, rs476406737, rs454104745 in *BOLA-DRB3*. The polymorphic SNPs were found on two genes: *TLR4* (rs460053411) and *BOLA-DRB3* (rs42309897, rs208816121, rs110124025). Thereby, a total of 35 markers (monomorphic and polymorphic) were analysed from a total of 10,430 genotypes assayed. The average call rate for all successfully genotyped SNPs was 95.98%. A total of 8 alleles and 11 genotypes were found at 4 polymorphic markers in the 298 cattle.

The level of polymorphic SNPs was equal in both cattle breeds and of those polymorphic markers, only three (rs42309897, rs208816121 and rs110124025 in *BOLA-DRB3*) had a minor allele frequency ≥ 5% (MAF ≥ 0.05) (Figure 1) and one (rs460053411 in *TLR4*) was a rare SNP (MAF < 0.01) and was removed from all further analyses. The graphical KASP results displayed three clusters, in addition to the non-template controls, corresponding to the two homozygous samples and to heterozygous samples. For the three SNPs (rs42309897, rs208816121 and rs110124025 SNP) in the *BOLA-DRB3* gene, grouping of the samples according to the genotypes was observed (Figure 1A–C).

The genotype and allele frequencies for the polymorphic SNPs in Romanian Spotted and Romanian Brown cattle are shown in Table 3. Within the breeds, a higher frequency of the minor alleles (0.35 and 0.32) was recorded in the Brown breed compared to the Spotted (0.14 and 0.06) for rs42309897 and rs208816121, respectively. The frequency of minor allele for rs110124025 was similar in the Romanian Spotted and Romanian Brown breeds (0.26 and 0.21, respectively).

The chi-square test (*χ^2^*) test showed that the populations were in Hardy–Weinberg equilibrium for all SNPs in the *BOLA-DRB3* gene, except for rs208816121 (*p* < 0.05) in Romanian Spotted and rs460053411 in the *TLR4* gene that was removed from all further analyses (Table 3). The gene diversity of a marker locus (also called expected heterozygosity) and PIC values represent measures of genetic diversity among genotypes in populations, being primary measures regarding genetic variation in a population [28]. The genetic diversity indices of Ho, He and PIC were calculated and are summarized in Table 3.

The expected and the observed heterozygosity values obtained in the two investigated cattle breeds for *BOLA-DRB3* SNPs ranged from 0.116 to 0.459 and 0.008 to 0.604, respectively. The highest observed heterozygosity, which indicates an increased within-population diversity, was achieved for rs208816121 in the Romanian Brown population, with a value of 0.604. However, the average observed heterozygosity was less than the expected heterozygosity. The mean expected heterozygosity in the two breeds was 0.245 and 0.411 while the mean observed heterozygosity varied from 0.063 in Romanian Spotted to 0.243 in Romanian Brown. As the observed heterozygosity was lower than expected, we hypothesized that such discrepancies might be the result of inbreeding within the population. The observed heterozygosity also indicates low to medium levels of genetic diversity in the studied cattle breeds.

The maximum PIC value for *BOLA-DRB3* SNPs was 0.645 for rs42309897 in the Romanian Brown breed and the minimum PIC value was 0.232 for rs208816121 in the Romanian Spotted breed. Generally, our results showed that the PIC values for all the SNPs in the *BOLA-DRB3* gene in the Romanian Brown cattle were higher than 0.5, with an average value of 0.595, in which case it is considered that all markers are highly informative. Lower PIC values were obtained in the Romanian Spotted cows, with an average value of 0.400, suggesting that those SNPs are moderately informative markers.

The SNP genotypes for the *BOLA-DRB3* gene were analysed for association with mastitis in the Romanian Spotted and Romanian Brown breeds (Table 4).

Two traits were considered for mastitis, as follows: the frequency (how many times the disease occurred during the lactation, once or twice), as well as how many quarters were affected (one, two or four). The results showed that the most significant effects for frequency of mastitis and number of affected quarters were found in the case of rs110124025 in the Romanian Spotted (*p* = 0.0018 and *p* = 0.0051) and Romanian Brown (*p* = 0.0359 and *p* = 0.0500) breeds. A higher frequency of mastitis (occurring once) for the *AA* genotype was observed in both the Romanian Spotted and Romanian Brown cattle (29.3%, 37.5%) compared to the *GG* genotype (10.3%, 8.3%).

In the case of the number of affected quarters, the frequency for two and/or four quarters affected by mastitis was significantly higher in animals with the *AA* genotype compared to those with *GG* genotype, in both breeds. Thus, in the Romanian Spotted breed, the frequency of mastitis in two quarters was fourfold higher in the *AA* compared to the *GG* genotype (5.1% and 1.1%, respectively), while the frequency for mastitis affecting all four quarters of the udder was more than two-fold higher (5.1% and 2.2%, respectively). In Romanian Brown cows, the frequency for mastitis affecting two quarters of the udder was almost 9 times higher in *AA* compared to *GG* animals (25% and 2.8%, respectively), while the frequency affecting all four quarters was 4.5 times higher (12.5% and 2.8%, respectively).

These results indicate that an individual carrying the *AA* genotype for rs110124025 has a significantly higher risk of developing mastitis during lactation, as well as having significantly more quarters affected by mastitis compared to individuals carrying the *GG* genotype. No significant associations with mastitis (*p* > 0.05) were found for the other two polymorphic markers in the *BOLA-DRB3* gene, rs42309897 and rs208816121.

## 4. Discussion

The genetic architecture for resistance to mastitis in dairy cattle constitutes a great interest for current research, particularly in the last decade, given the intense selection and use of sires with high breeding value to maximize milk yield, which has had detrimental effects on functional traits such as udder health. The goal of the current research was to detect the genetic background involved in resistance to mastitis through analysis of SNP markers belonging to genes that were found to be associated with genetic resistance to mastitis, and evaluate the usefulness of the selected markers in Romanian Spotted and Romanian Brown cattle breeds. While considering that clinical mastitis shows low heritability [29,30], the current research can contribute to the identification of SNP markers for future selection programs against mastitis.

The analyses were performed in Romanian Spotted and Romanian Brown cattle breeds on SNPs located in six genes (*CXCR2, CXCL8, BRCA1, LTF, BOLA-DRB3* and *TLR4*) using Kompetitive Allele Specific PCR technology. Among the 89 SNPs, 54 (60.7%) failed and 35 (39.3%) were successfully genotyped, of which 31 SNPs were fixed. The KASP assay success rates in the present study were lower than in the literature, where success rates of 78.5% [31], 80.9% [32] and 94.8% [33] were previously achieved. Since the KASP genotyping method allows discrimination of only two alleles at a single SNP site, it is highly possible that the low success rate in our Romanian native breeds was due to the absence of the SNP alleles. Moreover, the fact that a large number of monomorphic markers were identified in our native Romanian dairy cattle breeds that were formed as a result of long-time, non-systematic crossings between Romanian Grey cows (belonging to Podolian cattle) and bulls of different breeds reveals that the breeds we investigated have a different genetic architecture for the markers included in our study compared to similar breeds from other countries.

Four of the investigated markers were polymorphic. Among polymorphic loci with MAF ≥ 5%, only the Romanian Spotted breed was in Hardy–Weinberg disequilibrium at the g.25475692C>T locus (rs208816121, *p* < 0.05). The rs110124025 SNP in the *BOLA-DRB3* gene had a significant effect on mastitis prevalence. The genetic variant *GA* of rs110124025 was at a low frequency in both breeds. This might be attributed to the relatively small number of animals studied. The homozygous genotypes were more frequent than the heterozygous genotypes for the other polymorphic markers in the *BOLA-DRB3* gene.

Results of our SNP marker analyses are in accordance with previous studies, which found that different *BOLA-DRB3* alleles are associated with somatic cell scores in Brazilian Gyr dairy cattle [34], variations in clinical mastitis in Norwegian Red cows [35] and susceptibility to infectious diseases such as tick resistance [36] or the bovine leukaemia virus [37]. Similar studies in Holstein cows have shown associations between alleles of *BOLA-DRB3.2* and various aspects of mastitis. For example, three alleles denoted *22, *23 and *24 were associated with mastitis caused by *Staphylococcus spp.* [38] and were resistant to mastitis [39]. In addition, alleles *01 and *52 were associated with the occurrence of clinical mastitis [40].

The present study highlights that the *BOLA-DRB3* gene represents an important locus affecting resistance to mastitis in cattle. The major histocompatibility complex (MHC) in cattle, also known as bovine leukocyte antigen (BOLA), was found to play a critical role in the immune system, and due to its primary function in pathogen identification, has been used as a disease marker in cattle [17]. *BOLA-DRB3* is a highly polymorphic gene and has been studied as a potential candidate locus for mastitis susceptibility. This gene is located in the most polymorphic bovine MHC region, with more than 130 genetic variants [17]. It plays an important role in the response to infectious agents being responsible for the differences in the immune activity of cattle. The *BOLA-DRB3* gene has been linked with resistance to different infectious diseases in cattle [35,36,37,41,42].

Current results agree with those published by Cziszter et al. [21] in a comparative study of the two breeds, Romanian Spotted and Romanian Brown, for their productive and functional traits, with some significant breed disparities being reported for both milk yield and health.

Data regarding the associations of *BOLA-DRB3* genes with mastitis resistance in Romanian cattle are quite limited. The rs110124025 SNP in the *BOLA-DRB3* was confirmed to be associated with mastitis resistance in our study. Therefore, our findings validate this SNP as a potential candidate for selection for udder health in Romanian dairy cattle.

## 5. Conclusions

The Kompetitive Allele Specific PCR genotyping assay was used to identify SNP markers in already known candidate genes associated with genetic resistance or susceptibility to mastitis. We have analysed the missense SNPs belonging to six genes (*CXCR2, CXCL8, TLR4, BRCA1, LTF, BOLA-DRB3*) and identified one marker (rs110124025) in *BOLA-DRB3* that was associated with frequency of mastitis and with the number of affected quarters. Our findings support the use of *BOLA-DRB3* alleles in future research on cattle genetic resistance for mastitis, as well as use as a candidate for future genomic selection programs. The current work contributes to an increase in knowledge regarding the genetic variability of SNPs for udder health in cattle.

## Figures and Tables

**Figure 1 animals-13-01484-f001:**
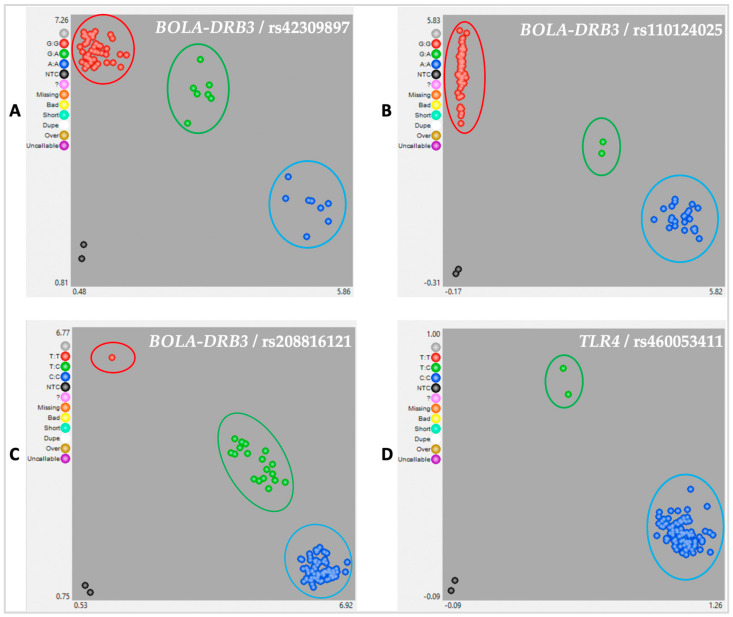
Graphics showing genotypes of the rs42309897 (**A**) rs110124025, (**B**) rs208816121, (**C**) SNPs in *BOLA-DRB3* gene and rs460053411 and (**D**) SNP in TLR4 gene, as well as a grouping of the data set. Red and blue dots correspond to homozygous genotypes, green dots correspond to heterozygous genotypes and black dots correspond to non-template control (NTC) samples.

**Table 1 animals-13-01484-t001:** Details of genes, chromosome location and genomic location of the 89 studied SNPs.

SNP ID	Gene Symbol	Chr	Position ^1^	Alleles	Remarks	Substitution/Position in Coding Region/Change in Amino Acid
rs478194554	*CXCR2*	2	106,191,506	A/T	missense	c.73A>T, p.Asn25Tyr
rs480434223	*CXCR2*	2	106,191,772	G/T	missense	c.339G>T, p.Lys113Asn
rs468701549	*CXCR2*	2	106,191,779	G/A	missense	c.346G>A, p.Val116Ile
rs209319366	*CXCR2*	2	106,192,040	A/C	missense	c.607A>C, p.Met203Leu
rs464576793	*CXCR2*	2	106,192,155	T/G	missense	c.722T>G, p.Leu241Arg
rs433132004	*CXCR2*	2	106,192,157	T/C	missense	c.724T>C, p.Phe242Leu
rs467207447	*CXCR2*	2	106,192,187	G/T	missense	c.754G>T, p.Ala252Ser
rs455755453	*CXCR2*	2	106,192,194	G/T	missense	c.761G>T, p.Arg254Leu
rs474074698	*CXCR2*	2	106,192,265	A/C	missense	c.832A>C, p.Thr278Pro
rs446793067	*CXCR2*	2	106,192,320	G/C	missense	c.887G>C, p.Gly296Ala
rs381099940	*CXCR2*	2	106,192,422	A/G	missense	c.989A>G, p.Lys330Arg
rs384398669	*CXCR2*	2	106,192,437	A/G	missense	c.1004A>G, p.His335Arg
rs134178216	*IL8 (CXCL8)*	6	88,810,896	T/G	missense	c.20T>G, p.Val7Gly
rs468955241	*IL8 (CXCL8)*	6	88,812,530	A/C	missense	c.81A>C, p.Arg27Ser
rs436292881	*IL8 (CXCL8)*	6	88,812,541	A/T	missense	c.92A>T, Glu31Val
rs448378725	*IL8 (CXCL8)*	6	88,812,575	A/C	synonymous	c.126A>C, p.Thr42=
rs466813222	*IL8 (CXCL8)*	6	88,812,578	T/C	synonymous	c.129T>C, p.Pro43=
rs433737959	*IL8 (CXCL8)*	6	88,812,616	A/T	missense	c.167A>T, p.Glu56Val
rs453466519	*IL8 (CXCL8)*	6	88,812,623	G/T	synonymous	c.174G>T, p.Gly58=
rs452742998	*IL8 (CXCL8)*	6	88,812,944	C/A	missense	c.222C>A, p.Asn74Lys
rs445886432	*IL8 (CXCL8)*	6	88,812,994	T/G	missense	c.272T>G, p.Val91Gly
rs8193041	*TLR4*	8	107,058,305	C/T	missense	c.10C>T, p.Arg4Cys
rs460053411	*TLR4*	8	107,058,332	C/T	missense	c.37C>T, p.Pro13Ser
rs8193048	*TLR4*	8	107,062,990	G/A	missense	c.148G>A, p.Asp50Asn
rs8193049	*TLR4*	8	107,066,042	A/C	missense	c.452A>C, p.Asn151Thr
rs8193050	*TLR4*	8	107,066,304	C/G	missense	c.714C>G, p.Asn238Lys
rs8193053	*TLR4*	8	107,066,630	C/A	missense	c.1040C>A, p.Ala347Glu
rs8193055	*TLR4*	8	107,066,732	A/G	missense	c.1142A>G, p.Lys381Arg
rs135659119	*TLR4*	8	107,067,191	C/G	missense	c.1601C>G, p.Ser534Ter
rs8193066	*TLR4*	8	107,067,538	G/A	missense	c.1948G>A, p.Val650Ile
rs437880924	*BRCA1*	19	43,070,163	A/G	missense	c.5543T>C, p.Val1848Ala
rs457333759	*BRCA1*	19	43,070,167	G/T	missense	c.5539C>A, p.Leu1847Met
rs470945089	*BRCA1*	19	43,070,173	T/G	missense	c.5533A>C, p.Thr1845Pro
rs439749719	*BRCA1*	19	43,070,175	T/G	missense	c.5531A>C, p.Asp1844Ala
rs453404290	*BRCA1*	19	43,070,180	C/G	missense	c.5526G>C, p.Glu1842Asp
rs473473146	*BRCA1*	19	43,070,181	T/C	missense	c.5525A>G, p.Glu1842Gly
rs442243059	*BRCA1*	19	43,070,189	C/A	missense	c.5517G>T, p.Gln1839His
rs462317177	*BRCA1*	19	43,070,190	T/G	missense	c.5516A>C, p.Gln1839Pro
rs482618972	*BRCA1*	19	43,070,194	A/C	missense	c.5512T>G, p.Tyr1838Asp
rs457592746	*BRCA1*	19	43,070,199	G/T	missense	c.5507C>A, p.Ala1836Asp
rs477576896	*BRCA1*	19	43,070,207	G/C	missense	c.5499C>G, p.Asp1833Glu
rs466537670	*BRCA1*	19	43,070,214	A/C	missense	c.5492T>G, p.Val1831Gly
rs480362322	*BRCA1*	19	43,070,229	A/C	missense	c.5477T>G, p.Val1826Gly
rs208338549	*BRCA1*	19	43,070,233	C/T	missense	c.5473G>A, p.Val1825Met
rs469103334	*BRCA1*	19	43,070,236	G/C	missense	c.5470C>G, p.Pro1824Ala
rs437904779	*BRCA1*	19	43,070,244	C/G	missense	c.5462G>C, p.Cys1821Ser
rs451586626	*BRCA1*	19	43,070,247	A/C	missense	c.5459T>G, p.Met1820Arg
rs456463862	*BRCA1*	19	43,070,254	C/T	missense	c.5452G>A, p.Gly1818Arg
rs384176726	*LTF*	22	52,953,375	A/G	missense	c.5A>G, p.Lys2Arg
rs461137889	*LTF*	22	52,953,380	T/C	missense	c.10T>C, p.Phe4Leu
rs483118032	*LTF*	22	52,953,384	T/G	missense	c.14T>G, p.Val5Gly
rs450733501	*LTF*	22	52,953,387	C/T	missense	c.17C>T, p.Pro6Leu
rs478177296	*LTF*	22	52,953,392	C/G	missense	c.22C>G, p.Leu8Val
rs445052454	*LTF*	22	52,953,398	T/G	missense	c.28T>G, p.Ser10Ala
rs478065524	*LTF*	22	52,957,166	G/A	missense	c.55G>A, p.Ala19Thr
rs460348489	*LTF*	22	52,957,271	A/C	missense	c.160A>C, p.Thr54Pro
rs453982323	*LTF*	22	52,960,346	G/T	missense	c.260G>T, p.Gly87Val
rs472508993	*LTF*	22	52,960,352	A/G	missense	c.266A>G, p.Asp89Gly
rs461145277	*LTF*	22	52,960,367	G/T	missense	c.281G>T, p.Arg94Leu
rs482729559	*LTF*	22	52,960,370	C/G	missense	c.284C>G, p.Pro95Arg
rs476824727	*LTF*	22	52,960,393	A/G	missense	c.307A>G, p.Thr103Ala
rs447239896	*LTF*	22	52,960,402	T/G	missense	c.316T>G, p.Ser106Ala
rs433547801	*BOLA-DRB3*	23	25,730,098	A/T	missense	c.104A>T, p.His35Leu
rs208667846	*BOLA-DRB3*	23	25,730,106	G/C	missense	c.112G>C, p.Glu38Gln
rs110364925	*BOLA-DRB3*	23	25,730,113	A/C	missense	c.119A>C, p.Tyr40Ser
rs137786474	*BOLA-DRB3*	23	25,730,116	A/C	missense	c.122A>C, p.Lys41Thr
rs210718478	*BOLA-DRB3*	23	25,730,118	A/G	missense	c.124A>G, p.Arg42Gly
rs461191302	*BOLA-DRB3*	23	25,730,119	G/A	missense	c.125G>A, p.Arg42Lys
rs379932839	*BOLA-DRB3*	23	25,730,151	G/T	missense	c.157G>T, p.Val53Leu
rs42313447	*BOLA-DRB3*	23	25,730,159	C/G	missense	c.165C>G, p.Phe55Leu
rs448858468	*BOLA-DRB3*	23	25,730,163	G/C	missense	c.169G>C, p.Asp57His
rs467367584	*BOLA-DRB3*	23	25,730,165	C/G	missense	c.171C>G, p.Asp57Glu
rs209865172	*BOLA-DRB3*	23	25,730,169	T/C	missense	c.175T>C, p.Tyr59His
rs465057416	*BOLA-DRB3*	23	25,730,173	T/A	missense	c.179T>A, p.Phe60Tyr
rs467605669	*BOLA-DRB3*	23	25,730,176	A/C	missense	c.182A>C, p.His61Pro
rs109928428	*BOLA-DRB3*	23	25,730,191	A/C/	missense	c.197A>C, p.Tyr66Ser
rs436108988	*BOLA-DRB3*	23	25,730,194	T/C	missense	c.200T>C, p.Val67Ala
rs476406737	*BOLA-DRB3*	23	25,730,194	A/G	missense	c.200T>C, p.Val67Ala
rs42309897	*BOLA-DRB3*	23	25,730,215	G/A	missense	c.221G>A, p.Gly74Asp
rs136458736	*BOLA-DRB3*	23	25,730,239	T/A	missense	c.245T>A, p.Leu82Gln
rs133097997	*BOLA-DRB3*	23	25,730,248	G/C	missense	c.254G>C, p.Arg85Pro
rs434083272	*BOLA-DRB3*	23	25,732,613	G/A	missense	c.448G>A, p.Gly150Ser
rs209853925	*BOLA-DRB3*	23	25,732,628	C/A	missense	c.463C>A, p.His155Asn
rs454104745	*BOLA-DRB3*	23	25,732,631	A/G	missense	c.466A>G, p.Ile156Val
rs437148562	*BOLA-DRB3*	23	25,732,650	G/A	missense	c.485G>A, p.Arg162Gln
rs42312242	*BOLA-DRB3*	23	25,732,780	G/T	missense	c.615G>T, p.Glu205Asp
rs209467115	*BOLA-DRB3*	23	25,732,791	G/A	missense	c.626G>A, p.Arg209Gln
rs208816121	*BOLA-DRB3*	23	25,733,626	C/T	missense	c.718C>T, p.Leu240Phe
rs110124025	*BOLA-DRB3*	23	25,734,153	A/G	missense	c.767A>G, p.His256Arg

^1^ Position based on the ARS-UCD1.2 genome assembly of *Bos taurus*. Chr: chromosome.

**Table 2 animals-13-01484-t002:** Success ratio of the 89 SNPs investigated through KASP assay for 298 samples in Romanian Spotted and Romanian Brown cattle.

Status	Number	Success Ratio	SNP/Gene
Polymorphic	4	4.50%	rs460053411/TLR4, rs42309897/BOLA-DRB3, rs208816121/BOLA-DRB3, rs110124025/BOLA-DRB3
Monomorphic	31	34.83%	rs480434223/CXCR2, rs209319366/CXCR2, rs464576793/CXCR2, rs467207447/CXCR2, rs446793067/CXCR2, rs134178216/CXCL8, rs468955241/CXCL8, rs448378725/CXCL8, rs466813222/CXCL8, rs453466519/CXCL8, rs452742998/CXCL8, rs445886432/CXCL8, rs8193041/TLR4, rs8193048/TLR4, rs8193049/TLR4, rs8193050/TLR4, rs8193053/TLR4, rs8193055/TLR4, rs135659119/TLR4, rs8193066/TLR4, rs437880924/BRCA1, rs470945089/BRCA1, rs480362322/BRCA1, rs469103334/BRCA1, rs460348489/LTF, rs461145277/LTF, rs476824727/LTF, rs447239896/LTF, rs208667846/BOLA-DRB3, rs476406737/BOLA-DRB3, rs454104745/BOLA-DRB3
Failed	54	60.67%	rs478194554/CXCR2, rs468701549/CXCR2, rs433132004/CXCR2, rs455755453/CXCR2, rs474074698/CXCR2, rs381099940/CXCR2, rs384398669/CXCR2, rs436292881/CXCL8, rs433737959/CXCL8, rs457333759/BRCA1, rs439749719/BRCA1, rs453404290/BRCA1, rs473473146/BRCA1, rs442243059/BRCA1, rs462317177/BRCA1, rs482618972/BRCA1, rs457592746/BRCA1, rs477576896/BRCA1, rs466537670/BRCA1, rs208338549/BRCA1, rs437904779/BRCA1, rs451586626/BRCA1, rs456463862/BRCA1, rs384176726/LTF, rs461137889/LTF, rs483118032/LTF, rs450733501/LTF, rs478177296/LTF, rs445052454/LTF, rs478065524/LTF, rs453982323/LTF, rs472508993/LTF, rs482729559/LTF, rs433547801/BOLA-DRB3, rs110364925/BOLA-DRB3, rs137786474/BOLA-DRB3, rs210718478/BOLA-DRB3, rs461191302/BOLA-DRB3, rs379932839/BOLA-DRB3, rs42313447/BOLA-DRB3, rs448858468/BOLA-DRB3, rs467367584/BOLA-DRB3, rs209865172/BOLA-DRB3, rs465057416/BOLA-DRB3, rs467605669/BOLA-DRB3, rs109928428/BOLA-DRB3, rs436108988/BOLA-DRB3, rs136458736/BOLA-DRB3, rs133097997/BOLA-DRB3, rs434083272/BOLA-DRB3, rs209853925/BOLA-DRB3, rs437148562/BOLA-DRB3, rs42312242/BOLA-DRB3, rs209467115/BOLA-DRB3

**Table 3 animals-13-01484-t003:** Allele and genotype frequencies and main diversity indices [polymorphic information content (PIC), expected (He) and observed (Ho) heterozygosity] for the polymorphic SNPs in Romanian Spotted (RS) and Romanian Brown (RB) cows.

Gene/SNP ID	Breed	Genotype Frequency			Allele Frequency		PIC	He	Ho
*BOLA-DRB3* rs42309897		**GG**	**GA**	**AA**	**G**	**A**			
	RS	0.826 (*n* = 123)	0.074 (*n* = 11)	0.101 (*n* = 15)	0.86	0.14	0.400	0.238	0.073
	RB	0.611 (*n* = 22)	0.083 (*n* = 3)	0.306 (*n* = 11)	0.65	0.35	0.645	0.459	0.083
*BOLA-DRB3* rs208816121		**CC**	**TC**	**TT**	**C**	**T**			
	RS *	0.884 (*n* = 221)	0.108 (*n* = 27)	0.008 (*n* = 2)	0.94	0.06	0.232	0.116	0.108
	RB	0.375 (*n* = 18)	0.604 (*n* = 29)	0.021 (*n* = 1)	0.68	0.32	0.629	0.441	0.604
*BOLA-DRB3* rs110124025		**GG**	**GA**	**AA**	**G**	**A**			
	RS	0.740 (*n* = 185)	0.008 (*n* = 2)	0.252 (*n* = 63)	0.74	0.26	0.568	0.381	0.008
	RB	0.771 (*n* = 37)	0.042 (*n* = 2)	0.188 (*n* = 9)	0.79	0.21	0.511	0.333	0.041
*TLR4* rs460053411		**CC**	**TC**	**TT**	**C**	**T**			
	RS *	0.984 (*n* = 246)	0.016 (*n* = 4)	0.000 (*n* = 0)	0.99	0.01	0.046	0.015	0.016
	RB *	0.979 (*n* = 47)	0.021 (*n* = 1)	0.000 (*n* = 0)	0.99	0.01	0.059	0.020	0.020

* Not consistent with Hardy–Weinberg equilibrium.

**Table 4 animals-13-01484-t004:** The SNP genotypes and association with mastitis in the Romanian Spotted (RS) and Romanian Brown (RB) breeds.

Breed	SNP	Genotype	Mastitis Frequency (%)			Chi-Square d.f. *p* Value	Number of Affected Quarters (%)				Chi-Square d.f. *p* Value
			0	1	2		0	1	2	4	
RS	rs42309897	GG	85.1	13.2	1.7	2.836	85.1	9.6	1.8	3.5	7.774
		AA	93.3	6.7	0.0	4	93.3	0.0	6.7	0.0	4
		GA	72.7	27.3	0.0	0.5855	72.7	27.3	0.0	0.0	0.2551
	rs208816121	CC	85.4	13.1	1.5	0.500	85.4	9.2	1.9	3.5	1.383
		TC	88.9	11.1	0.0	2	88.9	7.4	3.7	0.0	2
		TT	-	-	-	0.7787	-	-	-	-	0.7094
	rs110124025	GG	88.6	10.3	1.1	12.611	88.6	8.1	1.1	2.2	12.791
		AA	69.0	29.3	1.7	2	69.0	20.8	5.1	5.1	2
		GA	-	-	-	0.0018	-	-	-	-	0.0051
RB	rs42309897	GG	85.0	5.0	10.0	3.856	85.0	5.0	5.0	5.0	5.415
		AA	81.8	18.2	0.0	4	81.8	9.1	9.1	0.0	4
		GA	66.7	33.3	0.0	0.4258	66.7	0.0	0.0	33.3	0.4918
	rs208816121	CC	82.2	11.8	5.9	0.152	82.3	5.9	5.9	5.9	0.923
		TC	85.7	10.7	3.6	2	85.7	7.1	3.6	3.6	2
		TT	-	-	-	0.9269	-	-	-	-	0.9612
	rs110124025	GG	88.9	8.3	2.8	6.654	88.9	5.5	2.8	2.8	7.775
		AA	50.0	37.5	12.5	2	50.0	12.5	25	12.5	2
		GA	-	-	-	0.0359	-	-	-	-	0.0500

d.f., degrees of freedom.; mastitis frequency: 0 = no mastitis, 1 = once, 2 = twice; number of affected quarters: 0 = no quarter, 1 = one quarter, 2 = two quarters; 4 = four quarters.

## Data Availability

The data presented in this study are available on request from the corresponding author. The data are not publicly available due to privacy reasons regarding the phenotypic data, which are owned by the Romanian Breeding Association “Bălțată Românească” Simmental type (ACVBR-SIM Harman-Brasov, Romania) and Innovative Agricultural Services (Reading, UK).

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
