# Peer review of "Kompetitive Allele Specific PCR Genotyping of 89 SNPs in Romanian Spotted and Romanian Brown Cattle Breeds and Their Association with Clinical Mastitis"

_animals, 2023, doi:10.3390/ani13091484_

Round 1
Reviewer 1 Report
The goal of this study was to find a way to improve animals' resistance to mastitis, a common and costly disease that affects dairy cattle. To do this, the authors investigated 298 cattle for 89 single nucleotide polymorphisms using Kompetitive Allele Specific PCR. This could help to identify cows with the most potentially protective immune responses. The results of the study showed that there is a significant association between a molecular marker in the BOLA-DRB3 gene and mastitis prevalence in the breeds investigated. This gene has been identified in previous studies as an important factor affecting resistance to mastitis in cattle.
Out of the 89 SNPs, only 35 SNPs were successfully genotyped. The authors need to explain and to discuss why so many SNPs were not successfully genotyped. Out of these 35 SNPs, only 3 were polymorphic with MAF>5% and thus, usable for association analysis. It's a pity that the authors were not able to perfrom association analysis on more than 3 SNPs. This and also the fact that these 3 SNPs belong only to one gene diminish the relevance of the study. Besides, the investigated breeds need to be introduced properly. A more detailed description of those breeds (relationship to other breeds, some characteristics) in the introduction would be desirable.
All in all, the manuscript is clearly written, and the data and results are accurately described and explained. I suggest to accept the manuscript after minor revision.
Further comments:
L37: p should be lowercase and italic. Please change this in the whole manuscript.
L54: Bos taurus should be italic.
L58-60: Provide citations for candidate genes and polymorphisms found in candidate genes.
L71: There should be a protected space between “receptor” and “4”, so that they are not separated by a new line.
L86: Use either a space or no space between BTA and the chromosome number. Please be consistent throughout the manuscript.
L145: Gene symbols in Table 1 should be written in italic. Further, commas should be used to make the genomic positions better readable -> 106,191,506
L170: The authors decided to use chi-square test for association analysis. Another possibility would be to use clinical mastitis as a binary trait and to use a generalized linear model. With this, the authors would be able to include fixed effects in the model which can help to control for sources of variation that are not directly related to the genotype. Please explain your decision.
L173: Please change to “clinical mastitis”
L180: It seems like the authors did not perform multiple testing correction. When multiple SNPs are tested using the chi-squared test or any other statistical test for genomic association, it is important to apply a multiple testing correction to control for the overall false positive rate or family-wise error rate (FWER). Please catch up for this.
L188: Why did so many SNPs fail? Please provide some explanation and add it to the results or discussion.
L212: Please add comma after gene, remove “the” before grouping and change the sentence to past tense, so that the sentence reads: “For the three SNPs in BOLA-DRB3 gene, grouping of the samples according to the genotypes were observed.“
L213-214: This sentence is repetition of previous sentence. Please combine both sentences.
L216: The resolution of the figure is not high enough. Please provide a figure with better resolution.
L226 Please remove the term “significantly” since this cannot be tested with a statistical test.
L233-239: Details of calculated measurements should be placed in the Material & Methods section. Please provide also the software with which those measurements were estimated.
L244: Table 2 should contain all 4 polymorphic SNPs.
L271: Dots have to be used as decimal separators. This is not the case for rs110124025 in RB.
L283: Dots have to be used as decimal separators of the p-values.
L284: Please remove the term “significantly” as this was not (and cannot be) tested with a statistical test.
Author Response
Dear reviewer, please find our point-by-point responses in the attached file

Reviewer 2 Report
Major Comments
(1) Concerns about MAF, see the flowing comments for Line 35-36 for details.
(2) In addition to chi-square to test whether the variant has an effect, could you also perform regression analysis to quantify the effect of each genotype or substitution effect of the minor allele?
(3) Low statistical power, instead of analysing variant one by one, could you compare with gene-based analysis (aggregate effect of multiple genic variants in a single test)?
(4) Given both Romanian Spotted and Romanian Brown are local breeds, please introduce more about the breeds and report whether their phenotypes (mastitis) differ from other well-known populations (such as Holsteins).
Minor comments:
Line 35-36: “The polymorphic SNPs with MAF<5% and call rate <95% were used for the association study.” I think the description here is wrong, since the association analyses should focus on “common” SNPs (MAF) with good genotyping quality (call rate)? Also, do you think the MAF threshold 5% is too strict? For example, Line 209, you mentioned you considered one variant with MAF <0.01 as “rare” and removed it from further analyses. I think the MAF close to 0.01 is still acceptable, especially when we assume variants with large effect could be less common because of negative selection.
Line 37, does this SNP remain statistically significant after correcting for false positives in multiple tests, e.g., Bonferroni correction?
Line 223-224, if the variant does not deviate from HWE, then this is something expected. Therefore, in my opinion, it should not be used as results.
Line 251, I think 3 digits are enough.
Line 278, the statement of “which had complete or incomplete variability” is not that clear for me.
Line 226-229, could you also check the allele frequency in other dairy cattle breeds such as Holsteins?
Author Response
Dear reviewer, please find attached a file with our responses to your comments

Reviewer 3 Report
The manuscript titled “Kompetitive Allele Specific PCR genotyping of 89 SNPs in Romanian Spotted and Romanian Brown cattle breeds and their association with clinical mastitis” reveal a significant association of a molecular marker in the BOLA-DRB3 gene with mastitis prevalence in the investigated breeds.
The manuscript is though well written and well presented, however, the study got several limitations. The manuscript is not acceptable in the present form and may be considered after major revisions.
Detail comments are as follows:
1. The abstract needs to be rephrased. Improve the materials and methods in the abstract and mention which statistical models are mentioned in the abstract
2. Give a details on the animal feeding and housing.
3. In 130th sentence 98 is written instead of 89 SNPs.
4. The authors should mention the level of SCC as well as what threshold was used to declare an animal mastitic.
5. Better to include the subclinical mastitis as well based on the SCC.
6. It is not mentioned how the mastitis confirmation was done? through SCC, California mastitis test or?
7. How about more than two mastitis?
8. Give detail note on the statistical model and package used in the materials and methods.
9. The language needs improvement.
Author Response
Dear reviewer, find our point-by-point responses in the attached file

Round 2
Reviewer 3 Report
Nill
Author Response
English language reviewed, and checked the spelling. See the final version of the manuscript uploaded